# Bioinformatics Analysis of the Molecular Networks Associated with the Amelioration of Aberrant Gene Expression by a Tyr–Trp Dipeptide in Brains Treated with the Amyloid-β Peptide

**DOI:** 10.3390/nu15122731

**Published:** 2023-06-13

**Authors:** Momoko Hamano, Takashi Ichinose, Tokio Yasuda, Tomoko Ishijima, Shinji Okada, Keiko Abe, Kosuke Tashiro, Shigeki Furuya

**Affiliations:** 1Department of Bioscience and Bioinformatics, Faculty of Computer Science and Systems Engineering, Kyushu Institute of Technology, 680-4 Kawazu, Iizuka 820-8502, Fukuoka, Japan; 2Laboratory of Functional Genomics and Metabolism, Faculty of Agriculture, Kyushu University, 744 Motooka, Nishi-ku, Fukuoka 819-0395, Fukuoka, Japan; 3Department of Bioscience and Biotechnology, Graduate School of Bioresource and Bioenvironmental Sciences, Kyushu University, 744 Motooka, Nishi-ku, Fukuoka 819-0395, Fukuoka, Japan; 4Graduate School of Agricultural and Life Science, The University of Tokyo, 1-1-1, Yayoi, Bunkyo-ku, Tokyo 113-8657, Japan; aishiji@mail.ecc.u-tokyo.ac.jp (T.I.);; 5Kanagawa Institute of Industrial Science and Technology (KISTEC), 3-25-13 Tonomachi, Kawasaki-ku, Kawasaki 210-0821, Kanagawa, Japan; 6Innovative Bio-Architecture Center, Faculty of Agriculture, Kyushu University, 744 Motooka, Nishi-ku, Fukuoka 819-0395, Fukuoka, Japan

**Keywords:** food peptide, Tyr–Trp, Alzheimer’s disease, working-memory deficits, Phgdh

## Abstract

Short-chain peptides derived from various protein sources have been shown to exhibit diverse bio-modulatory and health-promoting effects in animal experiments and human trials. We recently reported that the oral administration of the Tyr–Trp (YW) dipeptide to mice markedly enhances noradrenaline metabolism in the brain and ameliorates the working-memory deficits induced by the β-amyloid 25–35 peptide (Aβ_25–35_). In the current study, we performed multiple bioinformatics analyses of microarray data from Aβ_25–35_/YW-treated brains to determine the mechanism underlying the action of YW in the brain and to infer the molecular mechanisms and networks involved in the protective effect of YW in the brain. We found that YW not only reversed inflammation-related responses but also activated various molecular networks involving a transcriptional regulatory system, which is mediated by the CREB binding protein (CBP), EGR-family proteins, ELK1, and PPAR, and the calcium-signaling pathway, oxidative stress tolerance, and an enzyme involved in de novo l-serine synthesis in brains treated with Aβ_25–35_. This study revealed that YW has a neuroprotective effect against Aβ_25–35_ neuropathy, suggesting that YW is a new functional-food-material peptide.

## 1. Introduction

Peptides derived from food proteins have been shown to exhibit diverse bio-modulatory effects in animal and human studies [1,2]. Among the functions of such food-derived peptides, protective effects on the central nervous system have also been reported along with anti-inflammatory and anti-oxidative stress effects, improvement of cognitive function, antidepressant effects, and reductions in the symptoms of neurodegenerative diseases [3,4]. Although the mechanisms underlying the effects of these food-derived peptides vary greatly among tissue/cell types and remain largely unknown, it is assumed that they are mediated via interactions with endogenous target molecules that are present in human or animal tissues/cells.

The oral administration of a peptide mixture primarily composed of di- and tripeptides prepared from soy proteins via in vitro enzymatic digestion enhanced the synthesis and metabolism of catecholamines, especially noradrenaline, in the murine brain [5,6]. A similar effect was observed after the oral administration of a dipeptide, i.e., Ser-Tyr (SY), isolated from soy. Moreover, because the effect of SY was stronger than that of tyrosine alone [7], the Tyr-containing dipeptides, including SY, present in soy were assumed to be responsible for this observation. In fact, the soy peptide used in a previous study included a variety of Tyr-containing dipeptides [8].

To gain a mechanistic understanding of how Tyr-containing dipeptides act on the brain’s monoamine metabolism and behavioral phenotypes, we synthesized all possible Tyr-containing dipeptides and compared their effects on noradrenaline metabolism in the cerebral cortex of mice following oral administration. Compared with animals treated with SY, Tyr–Tyr, and other Tyr-containing dipeptides, the Tyr–Trp (YW)-treated group exhibited the highest content of 3-methoxy-4-hydroxyphenylethylene glycol, which is a noradrenaline metabolite, in the cerebral cortex [9]. Furthermore, our subsequent behavioral analysis demonstrated that the oral administration of YW protected against the working-memory deficits caused by the intracerebroventricular administration of a synthetic β-amyloid peptide (Aβ_25–35_), while also altering cortical gene expression [9]. Our preliminary gene expression analysis of the cerebral cortex in the Aβ_25–35_/YW-treated mice showed that approximately 54% of the genes that were significantly upregulated in the cerebral cortex of the Aβ_25–35_ group, including genes involved in chemokine and cytokine signaling pathways and addiction, were suppressed after the co-administration of YW. In contrast, YW administration restored approximately 62% of the genes that were significantly downregulated in the cerebral cortex of the Aβ_25–35_ group, including genes encoding proteins involved in protein synthesis and the dopaminergic synaptic pathway. These modifications may have contributed to the neuroprotective effects of YW against Aβ_25–35_ toxicity; however, the molecular mechanisms underlying the YW-induced neuroprotection against Aβ_25–35_ remain largely unknown.

Bioinformatics is a powerful research field with respect to analyzing vast and complex omics data, and bioinformatics approaches were used to elucidate complex molecular mechanisms in our previous studies [10,11]. In the central nervous system, a microarray analysis clarified the molecular mechanisms of the neurogenesis defects caused by *Phgdh* deletion [12] and the detection of crucial hub gene networks and molecular pathological mechanisms [13]. In addition, comprehensive analyses of gene-expression data have enabled the evaluation of the effect of chemicals and nutritional conditions on brain function [14,15]. Based on these findings, bioinformatics analyses of brain-based gene-expression data seem to be useful for the elucidation of the molecular mechanism underlying the effect of food peptide administration to the brain.

In the present study, we performed multiple bioinformatics analyses of the microarray data from Aβ_25–35_/YW-treated mouse brains and inferred the molecular network underlying the action of YW in the brain. Interestingly, we show herein that YW drove diverse molecular networks involving a transcriptional regulatory system mediated by the calcium-signaling pathway and oxidative stress tolerance in addition to various transcription factors. Furthermore, the present analysis suggests that YW administration ameliorated the expression level of the CREB binding protein (CBP) and an enzyme involved in the de novo synthesis of l-serine in Aβ_25–35_-treated brains.

## 2. Material and Methods

### 2.1. Ethics Approval

The microarray array data analyzed in this study were obtained in our recent study, which included animal experiments [7]. All of the animal experiments were conducted in accordance with the Standard Relating to the Care and Management of Laboratory Animals and Relief of Pain (Notice No. 88, Ministry of the Environment, Government of Japan). All experiments were reviewed and approved by the Animal Experiment Committee of Kyushu University (Permit No. A19-333) and Nihon Bioresearch, Inc. (Hashima, Japan) [7].

### 2.2. DNA Microarray Analysis and Data Acquisition

GEO-NCBI (GEO accession number: GSE146400) provided microarray data on the cerebral cortex (Cx) from male ddY mice (10 weeks of age) that had been subjected to intracerebroventricular injections of saline (*n* = 4), Aβ_25–35_ (3 μL, 6 nmol) (*n* = 5), and Aβ_25–35_ together with YW (100 mg kg^−1^/day, twice a day) (*n* = 5). Raw microarray signal intensity data (.CEL files) were log2-transformed and normalized via SST-RMA using a quantile algorithm [16] and the Thermo Fisher Expression Console™ 1.1 software(Thermo Fisher Scientific, Waltham, MA, USA). Subsequently, to identify differentially expressed genes, the data were analyzed using the Linear Models for Microarray Analysis (limma) package [17] of the Bioconductor software [18]. Significantly and differentially expressed genes were selected as follows: limma *p*-value < 0.05 and ratio > 1.0-fold (upregulated genes), and limma *p*-value < 0.05 and ratio < 1.0 (downregulated genes).

### 2.3. Gene Ontology (GO) and Kyoto Encyclopedia of Genes and Genomes (KEGG) Pathway Enrichment Analyses

For GO and KEGG pathway enrichment analyses of differentially expressed genes (DEGs), the Database for Annotation, Visualization, and Integrated Discovery (DAVID) [19] (https://david.ncifcrf.gov/, accessed on 30 November 2021) was used. The top three GO terms in the annotation clusters with statistical significance (*p* < 0.05) regarding the Functional Annotation Clustering function were extracted. DAVID was also used to calculate the enrichment *p*-values of all extracted GO terms for each module.

### 2.4. Gene Set Enrichment Analysis (GSEA)

A GSEA [20] was carried out using the GSEAPreranked program, as described previously [4]. Briefly, the program processed the “datasets” calculated from the sham and Aβ_25–35_ groups as well as the “gene sets” derived from the change in the gene-expression pattern between the Aβ_25–35_ and Aβ_25–35_+YW groups based on the microarray data. For the Aβ_25–35_ “dataset”, the fold change in gene expression (Aβ_25–35_/sham) necessary to generate a transcriptome-wide ranking list of gene expression was calculated. To create the ranking list, the gene symbols and log2-fold changes were sorted in decreasing order. Furthermore, the “gene sets” were defined as the 165 genes that were significantly upregulated in the Aβ_25–35_ group and suppressed in the Aβ_25–35_+YW group as well as the 156 genes that were significantly downregulated in the Aβ_25–35_ group and upregulated in the Aβ_25–35_+YW group, respectively.

### 2.5. Visualization of the Protein–Protein Interaction (PPI) Network and Module Detection

A PPI network was built using the PPI data stored in the Search Tool for the Retrieval of Interacting Genes (STRING) [21] Database (https://www.string-db.org/, accessed on 30 November 2021), in which PPIs are based on various types of evidence, such as experiments, databases, co-expression, neighborhood, gene fusion, and co-occurrence. The PPI network involving the DEGs was extracted. Moreover, Cytoscape was used to visualize the PPI network [22]. The degree of centrality was calculated by using Cytoscape to detect nodes with high centrality, which is indicated by the size of the node. The “combined score” of STRING was used to represent protein associations according to the thickness of the edge.

### 2.6. Ingenuity Pathways Analysis

As described previously, biologically relevant networks were built using the Ingenuity Pathways Analysis program (http://www.Ingenuity.com, accessed on 20 December 2021) [10,12], which develops functional molecular networks that overlay genes in a dataset based on algorithmically generated connectivity in gene–gene, gene–protein, and protein–protein interactions. The *p*-values for each network were calculated using this program by comparing the number of focus genes that were mapped in a given network to the total number of occurrences of those genes across all networks. The score for each network is presented as the negative log of the *p*-value, which indicates the probability of finding a set of genes in the network by chance.

### 2.7. Statistical Analyses

To detect DEGs in each group, the fold changes in gene expression in the Aβ_25–35_ group and the Aβ_25–35_+YW group were calculated and compared with those of the sham group and the Aβ_25–35_ group, respectively. An unpaired two-tailed Student’s *t*-test was used with an adjusted *p* < 0.05. All statistical analyses and graphical constructions were performed using R. To visualize the heatmap of the gene-expression levels, we used MeV (http://www.tm4.org/mev.html, accessed on 3 September 2021).

## 3. Results and Discussion

Previous studies confirmed that Aβ aggregates form near the sites of injection of Aβ into the mouse brain [23]. Concomitantly, a behavior test using the Morris water maze task demonstrated that mice that received the Aβ injection exhibited impairment of their learning/memory abilities. Our previous study also demonstrated that Alzheimer’s disease (AD) model mice that received a single injection of Aβ_25–35_ showed short-term memory impairment in the Y-maze test [9].

To elucidate the molecular mechanism underlying the effect of YW on the short-term memory impairment observed in mice treated with Aβ_25–35_, we reanalyzed the microarray data from the cerebral cortex of the sham, Aβ_25–35_, and Aβ_25–35_+YW groups [9]. To visualize the relationship between the samples from each group, we performed a principal component analysis (PCA) for dimension reduction in the gene-expression data. The clusters from the sham and Aβ_25–35_ groups were the furthest apart, whereas the cluster from the Aβ_25–35_+YW group was centered between the sham and Aβ_25–35_ groups (Figure 1A). Moreover, 1068 genes were upregulated and 1030 genes were downregulated by the Aβ_25–35_ treatment compared with the sham group (Figure 1B). In addition, when these DEGs were compared between the Aβ_25–35_ group and the Aβ_25–35_+YW group, 162 genes were upregulated while 167 genes were downregulated in the Aβ_25–35_+YW group (Figure 1C).

Among these altered genes, we identified 165 genes that were significantly upregulated in the Aβ_25–35_ group and downregulated in the Aβ_25–35_+YW group, whereas 156 genes were significantly downregulated in the Aβ_25–35_ group and upregulated in the Aβ_25–35_+YW group (Figure 2A,B, Appendix A). A GSEA confirmed that the 165 genes that were downregulated by YW in the Aβ_25–35_ group were significantly enriched in genes that were upregulated in the Aβ_25–35_ group (Figure 2C left), whereas the 156 genes that were upregulated by YW in the Aβ_25–35_ group were significantly enriched in genes that were downregulated in the Aβ_25–35_ group (Figure 2C, right).

To unveil the biological functions of the genes whose expression levels were significantly reversed by the YW treatment in the Aβ_25–35_-treated cerebral cortex, we performed GO and KEGG pathway enrichment analyses. The enrichment of GO terms associated with “transcription” among the 165 genes whose expression levels were significantly downregulated by YW in the Aβ_25–35_-responsive genes (Figure 3A) suggested that the YW treatment suppresses the Aβ_25–35_-induced transcriptional activation. Furthermore, the KEGG pathway associated with “calcium signaling” was significantly enriched in these 165 genes. The excessive accumulation of various Aβ peptide fragments has been shown to cause neuronal dysfunction via aberrant neuronal calcium homeostasis [24], suggesting that YW counteracts the dysregulated neuronal calcium homeostasis in the Aβ_25–35_ cerebral cortex. In turn, the GO terms “cellular oxidant detoxification” and “oxidation–reduction process” were enriched in the 156 genes whose expression levels were significantly upregulated by YW in the Aβ_25–35_-responsive genes. Furthermore, the KEGG pathway associated with “PPAR signaling” was significantly enriched among these 156 genes. The peroxisome proliferator-activated receptor-α (PPAR-α) regulates genes associated with neural glutamate homeostasis and cholinergic/dopaminergic signaling in the brain [25]. PPAR-β/δ and PPAR-γ, both part of the PPAR-α family, are expressed in the brain and other organs and regulate oxidative stress, energy homeostasis, fatty acid metabolism, and inflammation in mitochondria [26]. Since the expression levels of the *PPAR-α* and PPAR-γ co-activator 1α (*PGC-1α*) genes were drastically suppressed in AD brains, an attempt to restore their expression appeared to be a novel therapeutic target for improving aberrant neuronal metabolism and cognitive decline in the AD brain [25]. Taken together, these findings indicate that YW activated PPAR signaling in the cerebral cortex of the Aβ_25–35_-treated mice.

Next, using PPI, we performed a network analysis of the reversed genes detected after YW administration to identify the key factors whereby YW alleviates the working-memory deficit of Aβ_25–35_-treated mice. A node was defined as a protein encoded by genes whose was significantly reversed by YW administration, and its size indicated the score that was calculated using centrality analysis. This analysis identified CREBBP (CBP) as the most central node in the PPI network, with genes that were significantly downregulated after YW administration in the Aβ_25–35_-treated cerebral cortex (Figure 4A,C). CREBBP activation has been shown to ameliorate learning and memory deficits in AD brains by upregulating the brain-derived neurotrophic factor (BDNF) [27] through the regulation of various genes that bind to the cAMP-response element binding protein. Proteins involved in cell-cycle regulation, such as Cdk1, Cdkn1a, and Cdk4, were discovered near CREBBP (as well as Neurod2) by intermediating nuclear receptors such as Nr4a3 (Nor1) and Nr4a2 (Nurr1). The current PPI network analysis suggests that CREBBP is involved in the YW-induced downregulation of gene expression, which ameliorates the impairment of brain functions caused by Aβ_25–35_ administration.

Rps27 and Phgdh were identified as the central nodes in the PPI network consisting of genes significantly upregulated in the Aβ_25–35_-treated cerebral cortex after YW administration (Figure 4B,C). A previous study based on an in silico analysis approach reported that Rps27a is encoded by a hub gene in the AD brain [28,29] and carries out the regulation of microglial activity in neurodegenerative diseases [30]. Phgdh, which is the first committed enzyme that is necessary for de novo l-serine synthesis in the brain and other tissues [31,32], mediates neuron/cell survival and growth [10,31,33] as well as neuronal transmission by supplying l-serine and d-serine to the central nervous system [32]. Furthermore, a recent study demonstrated that supplementation with l-serine reverses the loss of NMDA-receptor-mediated long-term potentiation and spatial memory in the brains of AD model mice [22]. The PPI network analysis demonstrated that YW administration activated a network in which Rps27 and Phgdh may play critical roles in neuronal survival and/or function against Aβ_25–35_ toxicity.

To identify other master regulators that respond to YW administration, we performed an Ingenuity Pathway Analysis (IPA). As a result, three gene regulatory networks were found to be potentially involved in the YW-mediated amelioration of altered gene expression in the Aβ_25–35_-treated cerebral cortex, with ELK1, IL3, and amyloid precursor protein (APP) at the center of each network, respectively. In silico IPA demonstrated that the three networks, with ELK1, IL-3, and APP as the central regulators, respectively, contributed to the YW-induced changes in gene expression in the Aβ_25–35_-treated cerebral cortex. YW appeared to suppress the ELK1 and IL-3 networks and EGR-family transcription factors. ELK1 has been linked to neuronal cell death in various neurodegenerative diseases [34], whereas IL-3 is a multifunctional cytokine that has been linked to inflammatory and autoimmune diseases. In the brain, it is expressed in astrocytes and neurons and has been linked to neurodegeneration through its promoting the activation of microglia and macrophages [35]. In contrast, recent studies have shown that astrocyte-derived IL-3 programs microglia to accumulate and remove aggregates of Aβ and tau, thereby contributing to the prevention and improvement of the AD pathology [36]. The current IPA results showed that the inhibition of the IL3-centered network in the cerebral cortex of the YW-administered group affected the microglial biology in the Aβ_25–35_-treated cerebral cortex. It is worth noting that the YW treatment suppressed the ELK1 and IL-3 networks, both of which comprised genes from the early growth response (EGR) family, whose expression was also suppressed, whereas the suppressed IL-3 network was associated with the upregulation of Phgdh (Figure 5A,B). EGR-1 is a transcription factor that regulates cell survival, proliferation, differentiation, and neuronal functions such as memory, cognition, and synaptic plasticity [37] Moreover, EGR-1 inhibition is a potential therapeutic target for reducing inflammation and the progression of AD [38]. These results suggest that YW administration may induce the downregulation of transcriptional networks comprising the EGR family in the brain, which may then contribute to neuronal protection against Aβ_25–35_ toxicity. Moreover, the APP has also been identified as a regulatory center that is negatively connected to EGR1 and CREBBP (Figure 5C) and is encoded by a gene that is directly involved in the pathogenesis of AD when its proteolytic metabolism is dysregulated [39]. These IPA results suggest that the mechanism underlying the improvement of the behavioral deficit observed in the Aβ_25–35_ group after YW administration consists of the suppression of ELK1 and IL3 function, as well as EGR-1 expression, via the upregulation of APP (Appendix A).

Furthermore, we used an IPA to investigate the target of the canonical signaling pathway that responded to YW administration. Calcium ion signaling and several factors in G-protein-coupled-receptor (GPCR)-mediated nutrient sensing in the enteroendocrine cell pathway were activated in the Aβ_25–35_ group and downregulated in the Aβ_25–35_+YW group. In addition, cholecystokinin (CCK) is a neuropeptide that serves as a downstream target of calcium ion signaling in GPCR-mediated nutrient sensing in enteroendocrine cells, the expression of which was suppressed by YW administration (Appendix A). Therefore, we investigated the YW-induced changes in the expression of the cholecystokinin/gastrin-mediated signaling cascade in the cerebral cortex of mice treated with Aβ_25–35_. The results showed that the cholecystokinin-B-receptor-(CCKBR)-mediated Rho-associated coiled-coil-containing protein kinase (ROCK) pathway (Figure 6, left) and prostaglandin–endoperoxide synthase 2 (PTGS2) expression levels (Figure 6 bottom) were activated in the Aβ_25–35_-treated cerebral cortex, whereas these changes were attenuated in the Aβ_25–35_+YW group. ROCK inhibitors have been identified as a therapeutic pathway for reducing the accumulation of the pathogenic forms of the tau protein in tauopathies such as AD [40]. Furthermore, the inhibition of the ROCK signaling pathway ameliorated the cognitive deficits in the AD hippocampus by mitigating neuronal damage and apoptosis [41]. These results suggest that YW administration suppresses the activation of the ROCK signaling pathway, thereby ameliorating the working-memory deficit caused by Aβ_25–35_. CCK has been shown to trigger the expression of prostaglandin–endoperoxide synthase 2 (Ptgs2) [42], whereas its expression level was downregulated in the cerebral cortex of Aβ_25–35_+YW group compared with the Aβ_25–35_ group (Appendix A). The inflammatory response induced by Ptgs2 (cyclooxygenase, COX2) appears to complicate neurodegeneration in AD. Moreover, the suppression of the inflammatory response using anti-inflammatory drugs, such as nonsteroidal anti-inflammatory drugs (NSAIDs) and COX2 inhibitors, has been reported to be effective for the treatment and prevention of AD [43]. Taken together, these findings suggest that oral YW administration may be effective in ameliorating cognitive impairment in AD patients by suppressing PTGS2, potentially slowing the progression of inflammatory responses and subsequent neurodegeneration.

In this study, we elucidated the molecular mechanism underlying the improvement of short-term memory impairment in a mouse model of AD triggered by YW administration at the gene-expression level. YW peptide administration induced various gene-expression changes in the cerebral cortex of the AD model and ameliorated cognitive dysfunction via multiple pathways, including the modulation of calcium-signaling pathways, the enhancement of the NMDA receptor via the upregulation of Phgdh, and the suppression of inflammatory responses via the downregulation of EGR-1 and Ptgs2. Based on these results, a functional analysis of the amelioration of gene expression after YW administration will be necessary. In particular, further studies including genetic manipulation are needed to determine the functional involvement of individual genes in the pathways that respond to YW administration.

With the aging of the Japanese population, the prevention of the onset of AD and other age-related dementias and the alleviation of their symptoms are urgent issues with regard to the control of social healthcare costs, the maintenance of the healthcare system, and the preservation of the quality of life of the elderly and their families in the face of a declining working age population. To avoid an increase in the number of patients with dementia among the elderly, nutritional interventions that contribute to the maintenance of brain health in midlife must be developed along with the scientific basis for the effects of such foods. Previous studies of the effects of soy peptides on catecholamine neurotransmitter metabolism in the brain showed that their oral administration increased noradrenaline metabolite levels and metabolic turnover in the cerebral cortex, hippocampus, and brainstem [5,7]. The soy peptides used in these experiments consist of several Tyr-containing dipeptides [8]. The oral administration of a Tyr-containing dipeptide found in soy proteins, such as IY, SY, and YP, revealed that SY had the strongest effect on noradrenaline metabolism in the brain, which was also stronger than that observed for the same molar dose of Y alone. These results indicated that Tyr-containing dipeptides derived from food proteins may contribute more strongly to the maintenance and enhancement of brain noradrenaline metabolism compared with Y alone. Furthermore, these dipeptides have a different effect on the levels of brain noradrenaline hypermetabolism, suggesting that amino acid sequence specificity influences the effect on brain noradrenaline metabolism. In fact, a comparison of the effects of all Tyr-containing dipeptides, including SY, on noradrenergic metabolic turnover in the mouse brain showed that YW was the most potent molecule [9].

The YW dipeptide has been detected in food components and/or foods, including collagen [44] and sake/sake kasu [45]. In addition, YW appears in the primary sequences of plant-derived proteins. Thus, YW might be present in many types of plant-based peptides. In fact, our MS analysis confirmed the presence of YW in a commercially available soy peptide mixture (Appendix A). Conversely, milk whey protein can be a protein source of GTWY, which is a tetrapeptide that contains WY (the inverse sequence of YW), and it has been reported that its consumption improves working memory and novel object recognition in the Y-maze test [46]. Ano et al. also reported that the administration of the WY dipeptide alone reduces scopolamine-induced working-memory impairment [47]. In this study, WY promoted dopamine neurotransmission, whereas YW, which was used as a comparison peptide, did not alleviate the scopolamine-induced working-memory impairment. These results suggest that despite exhibiting reverse sequences, WY and YW have distinct effects on the central nervous system and that Tyr-containing dipeptides have diverse physiological activities in the brain. This indicates that these peptides employ different mechanisms for recognizing endogenous target molecules, which are most likely their sites of action in the brain. Peptides are gaining popularity as a new functional food material offering a high level of safety and a wide range of physiological functions because of their diverse amino acid sequence combinations [48]. However, their formational process, intestinal absorption, and pathways of transport into the brain are likely to be extremely diverse. In the future, the discovery of protein sources containing large amounts of the YW sequence that are capable of being converted into a dipeptide by digestive enzymes or microorganism-derived enzymes will be critical in the application of YW as brain food.

Multiple mechanisms cause cognitive dysfunction in patients with AD. The model generated via the intracerebroventricular injection of Aβ used in the present study was originally developed as an acute AD model that mimics sporadic AD, which accounts for the majority of all cases, and has emerged as a first-line model for the rapid screening of the symptom-relieving and neuroprotective effects of novel compounds and therapeutic strategies [49]. This model has been reported to exhibit impairment in cognitive functions, such as short-term memory, via the induction of neural damage responses in the brain, which include lipid peroxidation, protein nitrosylation, inducible nitric oxide synthase induction, and caspase-3 activation [50]. We have previously demonstrated the protective effect of YW administration on the short-term working-memory deficits induced in mice through a single intracerebroventricular injection of Aβ_25–35_ [9]. In the Aβ_25–35_ single-injection model of AD, non-spatial short-term working memory and spatial memory are impaired immediately after the administration of the peptide, whereas long-term memory is largely unaffected over time [51]. Since working and long-term memory deficits appear early in the progression of the central symptoms of AD, the Aβ_25–35_ injection model only partially reproduces the neurological symptoms of this disease. One of the limitations of this study is that it only estimated the molecular pathways underlying the protective effect of YW on working-memory deficits in a single animal model of the central symptoms of AD. Therefore, the efficacy of YW administration as a preventive and therapeutic treatment for multiple cognitive dysfunctions in patients with AD caused by various mechanisms remains unproven at this stage. To evaluate the validity of YW treatment and its underlying mechanism for improving the cognitive symptoms of AD, further studies are necessary using other animal models of this disease.

## 4. Conclusions

The current study revealed that the normalization of neurodegenerative gene-expression changes is a likely basis for the protective effect of YW against the neurotoxic effects of Aβ_25–35_, suggesting that YW is a potential impetus for drug discovery toward the prevention of age-related cognitive dysfunction.

## Figures and Tables

**Figure 1 nutrients-15-02731-f001:**
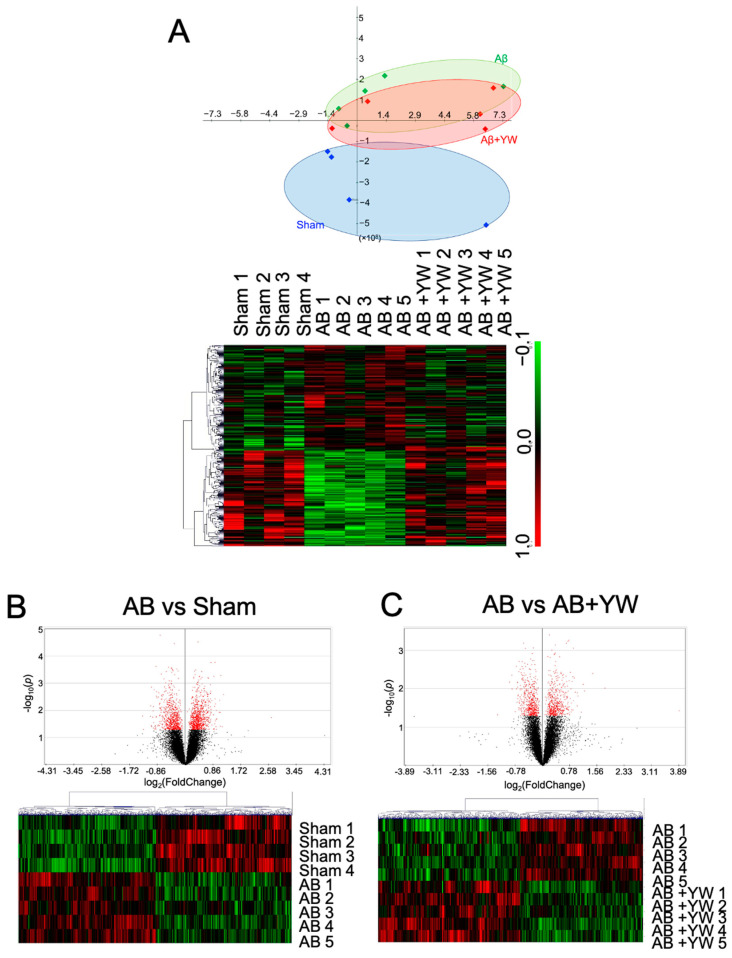
Alteration of the gene-expression profile caused by Aβ_25–35_+YW administration in the mouse cerebral cortex. (**A**) PCA plot of the sham, Aβ_25–35_, and Aβ_25–35_+YW groups based on gene-expression patterns. The blue dots indicate the sham group, the green dots indicate the Aβ_25–35_ group, and the red dots indicate the Aβ_25–35_+YW group. The heatmap shows the average expression levels of the genes that were significantly altered in each group. (**B**) Volcano plot of differential genes with altered expression between the Aβ_25–35_ group and the sham group (the black dots indicate all of measured genes, and the red dots indicate genes with a significantly altered expression level). The heatmap shows the average expression levels of the genes that were significantly altered in the Aβ_25–35_ group compared with the sham group. (**C**) Volcano plot of the differential genes with altered expression between the Aβ_25–35_+YW group and the Aβ_25–35_ group (the red dots indicate genes with a significantly altered expression level). The heatmap shows the average expression levels of the genes that were significantly altered in the Aβ_25–35_+YW group compared with the Aβ_25–35_ group.

**Figure 2 nutrients-15-02731-f002:**
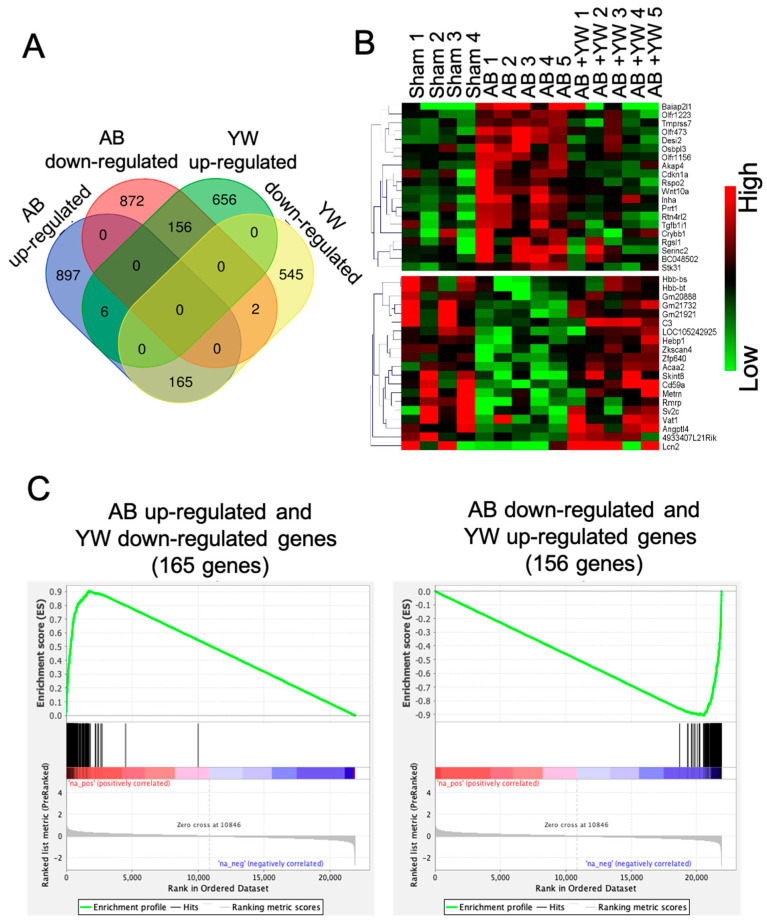
Extraction of genes significantly altered by YW administration in the brains of Aβ_25–35_-treated mice. (**A**) Venn diagram shows the number of genes significantly up-/downregulated in the Aβ_25–35_ group compared to the sham group and in the Aβ_25–35_+YW group compared to the Aβ_25–35_ group. (**B**) Heatmap of genes whose expressions were regressed in the Aβ_25–35_+YW group compared to the sham and Aβ_25–35_ groups. (**C**) Gene Set Enrichment Analysis (GSEA) for genes reversely expressed in the Aβ_25–35_+YW group among all significantly altered genes in the Aβ_25–35_ group. The left panel shows the enrichment of genes identified as significantly downregulated in the Aβ_25–35_+YW group, while the right panel shows the enrichment of genes significantly upregulated in Aβ_25–35_+YW group.

**Figure 3 nutrients-15-02731-f003:**
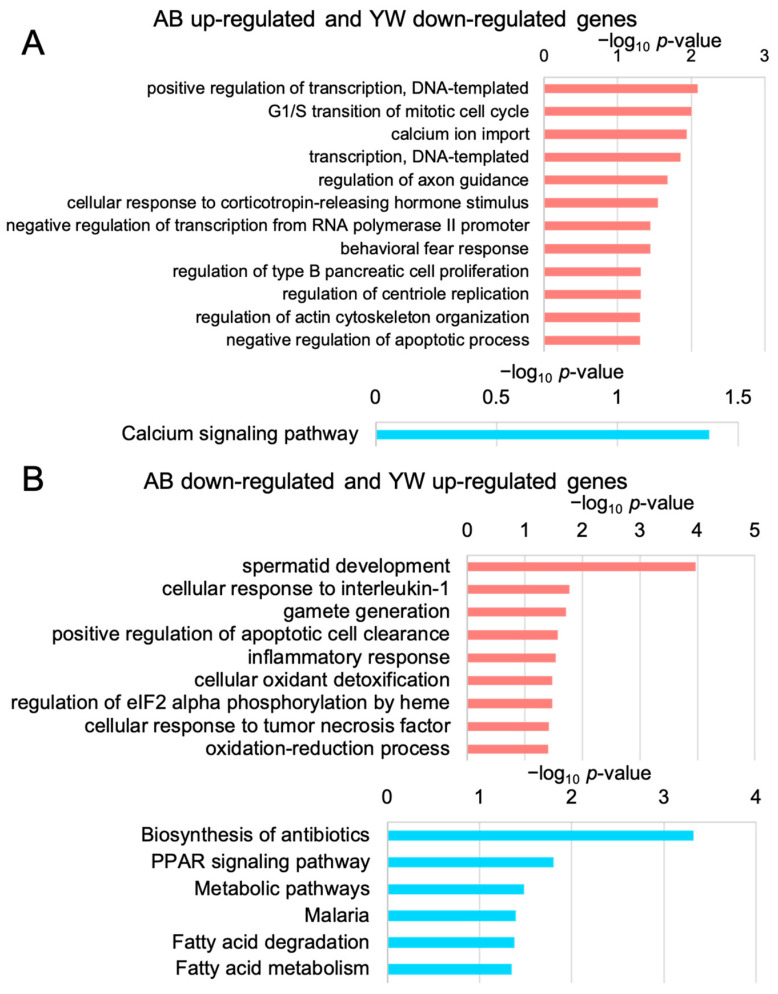
GO and KEGG pathway enrichment analyses of the genes that were significantly altered after YW administration. (**A**) GO (pink) and KEGG pathway (blue) enrichment analyses of genes that were upregulated in the Aβ_25–35_ group and downregulated in Aβ_25–35_+YW group. Low *p*-value (*p* < 0.05) terms are shown in each graph. The horizontal axis in each panel indicates −log_10_ *p*-values. (**B**) GO and KEGG pathway enrichment analyses of genes that were downregulated in the Aβ_25–35_ group and upregulated in the Aβ_25–35_+YW group.

**Figure 4 nutrients-15-02731-f004:**
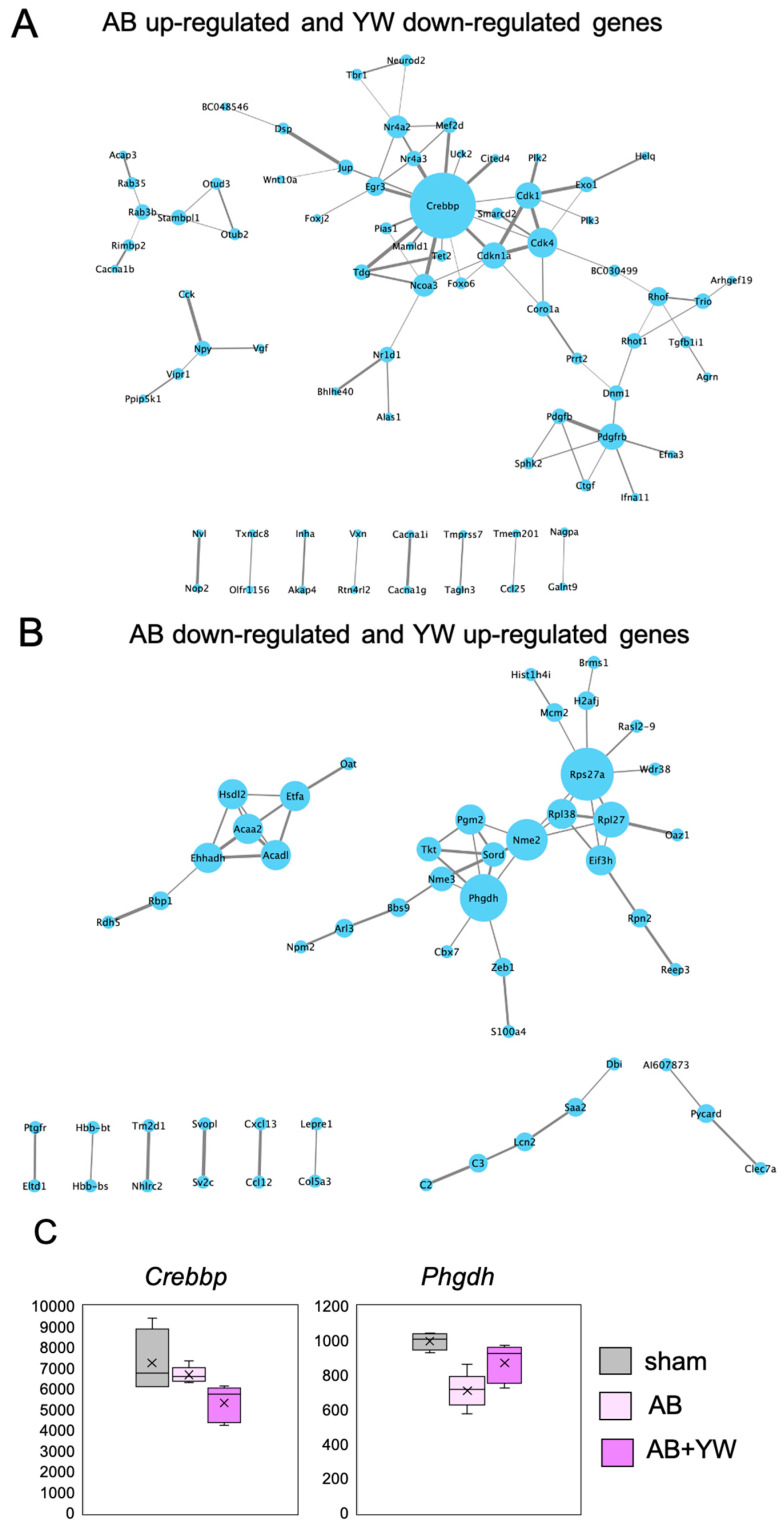
Protein–protein interaction (PPI) network analysis of genes for which activation was significantly elicited by YW administration. (**A**) PPI network of genes that were upregulated in the Aβ_25–35_ group and downregulated in the Aβ_25–35_+YW group. (**B**) PPI network of genes that were downregulated in the Aβ_25–35_ group and upregulated in the Aβ_25–35_+YW group. The nodes represent the proteins encoded by the genes with altered expression. The size of the nodes indicates the degree of centrality, and the thickness of the edges indicates the protein–protein association scores. (**C**) Boxplot showing the expression levels of the *Phgdh* and *Crebbp* genes, which were identified as hubs in the PPI network.

**Figure 5 nutrients-15-02731-f005:**
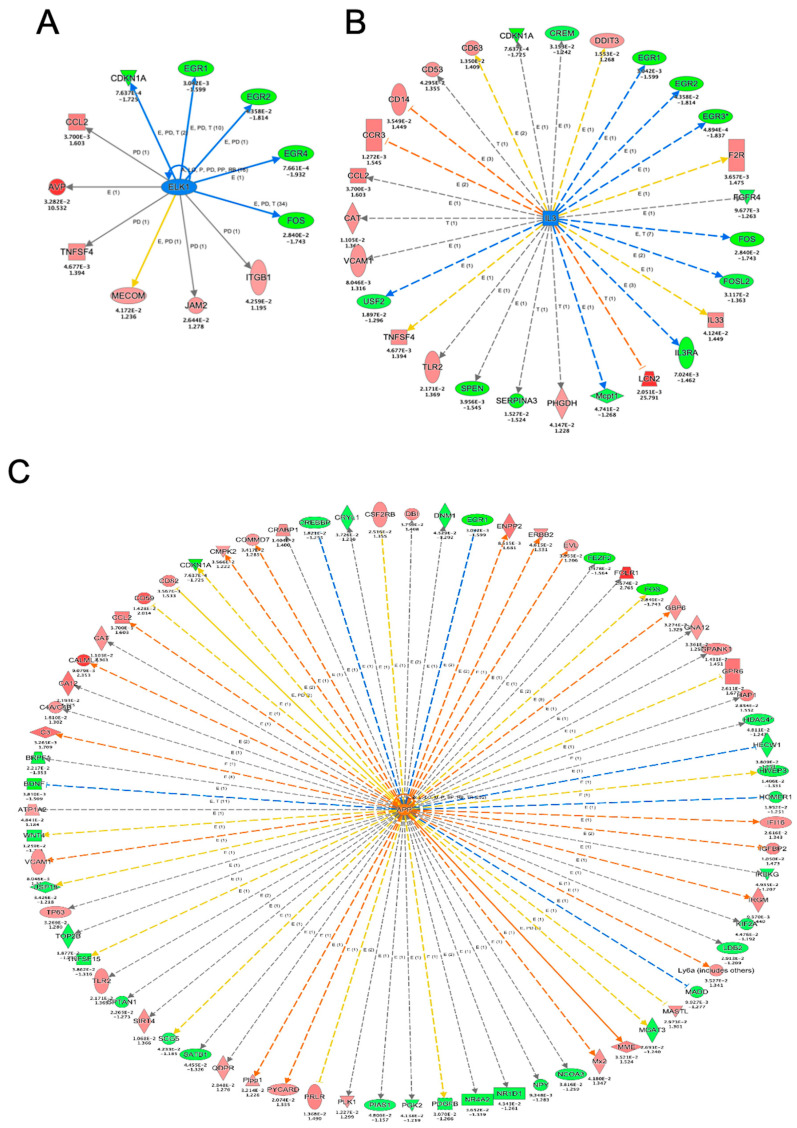
Ingenuity Pathway Analysis (IPA) for the detection of master regulator genes with altered expression after YW administration. Gene network centered on (**A**) ELK1, (**B**) IL3, and (**C**) APP, including genes with altered expression after YW administration compared with the Aβ_25–35_ group. The red nodes indicate upregulated genes, whereas the green nodes indicate downregulated genes. An asterisk in gene name indicates that multiple identifiers in the dataset file map to a single gene in the Global Molecular Network. (**D**) Heatmap showing the average expression level of the genes included in the ELK1-centered network. (**E**) Heatmap showing the average expression level of the genes included in the IL13-centered network. (**F**) Heatmap showing the average expression level of the genes included in the APP-centered network.

**Figure 6 nutrients-15-02731-f006:**
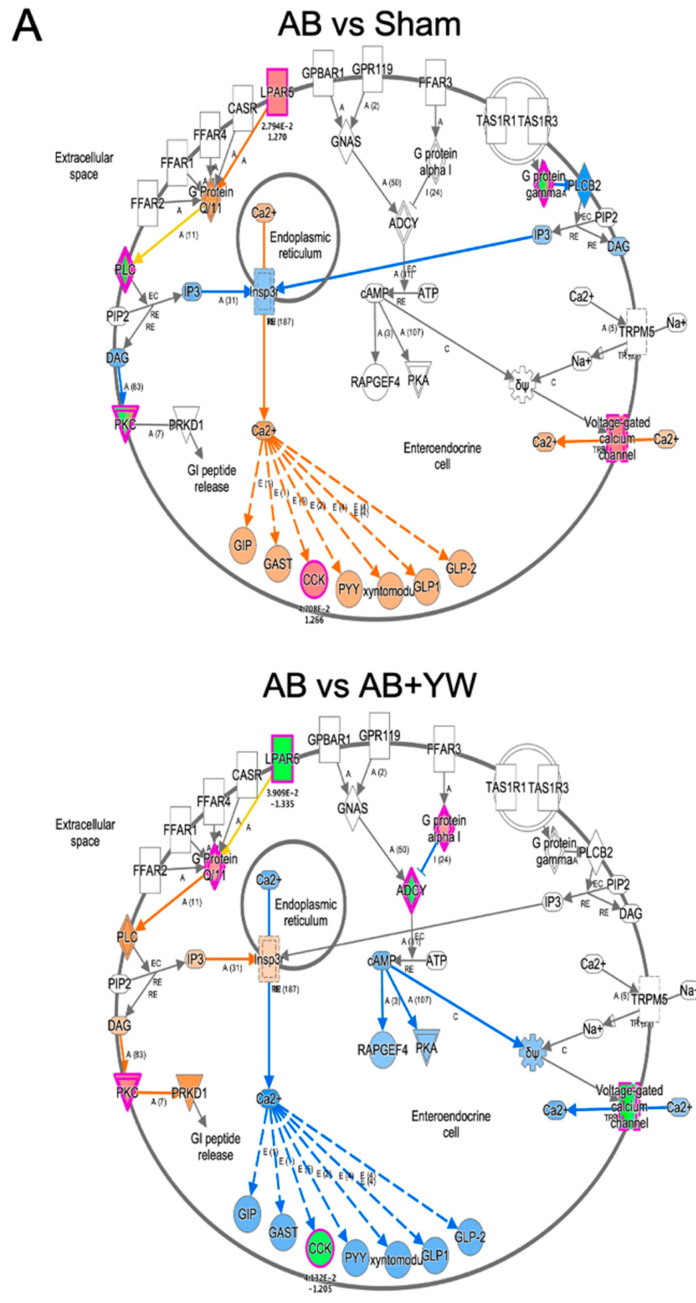
Detection of signaling pathways that were altered in the brain after YW administration. GPCR-mediated nutrient sensing in the enteroendocrine cell pathway (**A**), including calcium ion signaling, showing genes with altered expression in the Aβ_25–35_ group compared with the sham group (**top**) and in the Aβ_25–35_+YW group compared with the Aβ_25–35_ group (**bottom**). The orange nodes indicate upregulated genes, whereas the blue nodes indicate downregulated genes. Regarding cholecystokinin/gastrin-mediated signaling (**B**), genes with expression changes in the Aβ_25–35_ group were compared with the sham group (**top**) and genes with expression changes in the Aβ_25–35_+YW group (**bottom**). (**C**) Heatmap showing the average expression level of genes in the cholecystokinin-associated network. (**D**) Heatmap showing the average expression level of genes in the GPCR-mediated nutrient-sensing network.

## Data Availability

All microarray data were deposited in the National Center for Biotechnology Information Gene Expression Omnibus (http://www.ncbi.nlm.nih.gov/geo/; GEO Series accession number GSE 146400, accessed on 3 September 2021). The other data are shown in the paper and Appendix A.

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
