# Peer review of "Bioinformatics Analysis of the Molecular Networks Associated with the Amelioration of Aberrant Gene Expression by a Tyr–Trp Dipeptide in Brains Treated with the Amyloid-β Peptide"

_nutrients, 2023, doi:10.3390/nu15122731_

Round 1

Reviewer 1 Report

The manuscript is well written and highlights the important effect of the dipeptide Tyr-Trp in reducing abnormal gene expression in mouse cerebral cortex. The authors have done a good job explaining the mechanistic pathways through bioinformatics, however there are some key points which needs to be addressed:

1. As the manuscript mainly picks up its work from a previous study and there was no "oral administration" actually done in this study, I would suggest modifying the title a little bit as it may be misleading to readers who might think that the experiment was performed in this one. 

2. The animal model used in this study "ddY" is a model for IgA nephropathy. What is the rationale of using this animal model for unraveling the effects of memory deficits or AD? Mouse models such as the 5xFAD which are spontaneous models for AD pathogenesis would be a more appropriate choice to look at the effects of the dipeptide as the mechanistic pathways will be better represented. 

3. It might be beneficial to mention briefly the pathology visualized by injecting the ddY mice with Aβ25–35 so when we are discussing genes we know what phenotype we are looking for.

4. Why was cerebral cortex the only region analyzed in the study? It has been found that memory deficits and AD symptoms are more profound in the hippocampus region of the brain. It would be more appropriate to include this region to see the effects of Tyr-Trp.

5. The sample size of each group is not mentioned in the study.

6. The manuscript talks about food-derived peptides particularly soy, however, YW is not found in soy, it is one of the chemically synthesized peptide. Was the toxicity of the peptide assessed at the dose it is being administered? 

Author Response

8 March, 2023

Nutrients

Dear Editor:

Please find enclosed our revised manuscript. Following to reviewer’s comment, we revised the title from “Orally administrating Tyr-Trp dipeptide antagonizes aberrant gene expression in the mouse cerebral cortex treated with β-amyloid peptide fragments 25–35” to “Bioinformatic analysis of microarray data showing that Tyr-Trp dipeptide antagonizes aberrant gene expression in the mouse cerebral cortex treated with β-amyloid peptide fragment 25–35”.

We thank the editor and reviewers for the detailed and helpful comments. We revised the manuscript accordingly as indicated in our point-by-point responses detailed. We hope that you will find the revised manuscript suitable for publication in Nutrients.

Sincerely,

Momoko Hamano

Assistant professor

Department of Bioscience and Bioinformatics

Faculty of Computer Science and Systems Engineering

Kyushu Institute of Technology

Phone/FAX : +81-948-29-7831

E-mail : momoko@bio.kyutech.ac.jp

Shigeki Furuya

Professor

Laboratory of Functional Genomics and Metabolism

Graduate School of Bioresource and Bioenvironmental Sciences,

Kyushu University

Tel: +81-92-802-4741; Fax: +81-92-802-4741

E-mail: furuya.shigeki.805@m.kyushu-u.ac.jp

  1. As the manuscript mainly picks up its work from a previous study and there was no "oral administration" actually done in this study, I would suggest modifying the title a little bit as it may be misleading to readers who might think that the experiment was performed in this one.

Response:

Thanks for the comment regarding the title. We agree with the reviewer’s suggestion and have changed the title from “Orally administrating Tyr-Trp dipeptide antagonizes aberrant gene expression in the mouse cerebral cortex treated with β-amyloid peptide fragments 25–35” to “Bioinformatic analysis of microarray data showing that Tyr-Trp dipeptide antagonizes aberrant gene expression in the mouse cerebral cortex treated with β-amyloid peptide fragment 25–35”

  1. The animal model used in this study "ddY" is a model for IgA nephropathy. What is the rationale of using this animal model for unraveling the effects of memory deficits or AD? Mouse models such as the 5xFAD which are spontaneous models for AD pathogenesis would be a more appropriate choice to look at the effects of the dipeptide as the mechanistic pathways will be better represented.

Response:

The β-amyloid peptide (25-35) fragment forms aggregates in vitro as well as the (1-42) fragment and has been reported to cause neurotoxicity, amnesia, and cognitive dysfunction after in vivo intracerebroventricular administration (Kowall et al, Neurobiol Aging. 1992). Since then, this short amyloid peptide has been used as an inducer for cognitive deficits in experimental animals such as mice, rats, and monkeys. In 2004, Tohda et al. reported that memory impairment, axonal atrophy, and synaptic loss induced by intracerebroventricular administration of β-amyloid peptide (25-35) fragments to the ddY-strain were improved by oral administration of Kampo formula Zokumei-to (Tohda et al, Brain Research, 2003) and a saponin metabolite derived from ginseng (Tohda et al, Neuropsychopharmacology, 2004). Since then, the model injected by β-amyloid peptide (25-35) fragment to the ddY-strain has been used in many studies constantly.

In the present study, a combination model of β-amyloid peptide (25-35) fragments and ddY strain was employed in reference to these previous studies to evaluate the effects of orally administered dipeptides on cognitive function.

We are also aware of the usefulness of transgenic mice having forced expression of the gene mutations responsible for abnormal amyloid peptide accumulation and Tau accumulation in study of cognitive deficits as the reviewer pointed out, but these transgenic mouse models could not be not used in this study due to experimental limitations (P1A level laboratory and experimental size). Although there have been various animal models exhibiting cognitive impairment, no model has yet been reported that fully reproduces age-related cognitive dysfunction in humans. It is necessary to understand the disadvantages and advantages of each animal model and then select a model that meets the objectives of the research. Our present bioinformatics analysis has gained a new insight into a possible mechanism pertaining to the effect of oral YW administration to the β-amyloid peptide (25-35) fragment-ddY model on improving cognitive function. We believe it is worthwhile to publish in the journal.

  1. It might be beneficial to mention briefly the pathology visualized by injecting the ddY mice with Aβ25–35 so when we are discussing genes we know what phenotype we are looking for.

Response:

Thank you for your positive advice. Although the ddY mice we used in our experiments have not been assessed by such as immunostaining to visualize the pathology, previous studies have already reported the assessment of pathology and behavioral abnormality in the AD mice induced by similar methods. As you suggested, we have added the sentence at the beginning of the “3. Results and Discussion” section for the phenotype of Aβ25-35 injected into ddY mice as below.

Page 3, lines 136

Previous studies have confirmed that the Aβ aggregates formed near the sites of injection in the mouse brains by Aβ injection [19]. At the same time, a behavior test using the Morris water maze task demonstrated that mice receiving the Aβ injection were impaired in their learning/memory abilities. Our previous study also demonstrated that AD model mice have observed short-term memory impairment by Y-maze test [9].

  1. Why was cerebral cortex the only region analyzed in the study? It has been found that memory deficits and AD symptoms are more profound in the hippocampus region of the brain. It would be more appropriate to include this region to see the effects of Tyr-Trp.

Response:

As pointed out, previous studies have focused on hippocampus to elucidate the mechanisms of AD-induced memory impairment. In our previous study of the dipeptide administration, we have performed not only microarray but also metabolome analysis. These results showed that YW peptide markedly enhanced cerebral monoamine synthesis and metabolism in the cerebral cortex (Cx) [9]. Upon this result, we focused on Cx to clarify the molecular mechanism of the effects of YW administration. Aβ accumulation and pathological lesions have been observed not only hippocampus but also in the Cx of AD patients and AD model mice, and the Cx is one of the main regions of monoamine projection. From these findings, we determine that the analysis of Cx was considered appropriate for the evaluation of the effects of YW administration.

  1. The sample size of each group is not mentioned in the study.

Response:

As pointed out, the sample size of this study was not described. In this study, mice were divided for the experiment as Sham with n = 4, AB with n=5 and AB+YW with n=5. We have added the sample size to the method section of "DNA Microarray Analysis and Data Acquisition" as below.

Page 2, lines 79

GEO-NCBI (GEO accession number: GSE146400) provided microarray data of the cerebral cortex (Cx) from male ddY mice (10 weeks old) subjected to intracerebroventricular injections of saline (n = 4), Aβ25–35 (3 μL, 6 nmol) (n = 5), and Aβ25-35 with YW (100 mg kg-1 day-1, twice a day) (n = 5).

  1. The manuscript talks about food-derived peptides particularly soy, however, YW is not found in soy, it is one of the chemically synthesized peptide. Was the toxicity of the peptide assessed at the dose it is being administered?

Response:

Thank you for pointing this out. In our previous study, single administration of chemically synthesized dipeptide (1 mmol kg-1) didn’t show toxicity (Ichinose et al., Biosci. Biotechnol. Biochem., 2015). Therefore, ddy mice were administrated less than half amount of dipeptide twice a day in this study. No abnormal behavior was observed in 2 weeks administration, and the microarray analysis in the previous report suggested that YW improved excessive immune reactions (Ichinose et al., PLoS One, 2020). Thus, we believed that toxicity did not develop at this YW dose.

Taking your suggestion as an opportunity, we have updated our LC-MS method to analyze YW in soy peptide. As a result, although it was not in large amounts, YW was detected in soy peptide. In facts, YW appears multiple times in the primary sequence of glycinin and β-conglycinin witch are major soy storage proteins. We have also confirmed that YW appears in the primary sequences of various other plant-derived proteins. Therefore, we would like to modify the manuscript as bellow.

Page 6, lines 298

YW dipeptide has been detected in food components and/or foods, including collagen[39] and sake/sake kasu [40]. In addition, YW appears in the primary sequences of plant-derived proteins. Thus, YW might be contain in many kinds of plant-based peptides. On the other hand, milk whey protein can be a protein source for GTWY, a tetrapeptide containing WY (the inverse sequence of YW), and its consumption has been reported to improve working memory and novel object recognition in the Y-maze test [41].

Reviewer 2 Report

The author performed multiple bioinformatics analysis on the microarray data of A 25-35/YW-treated mouse brain, as well as inferred the molecular network of YW action in the brain. I am afraid I have to reject this manuscript at this time because I can not see any Figures in this manuscript so I can’t decide. And the supplementary files also do not include the figures. For these context I can see, please check the abstract, line the second last sentence of the abstract, “We found that YW can not only reverse inflammation-related responses but also activate diverse molecular networks involving transcriptional regulatory system mediated by CREBBP (CBP), EGR family, ELK1, and PPAR, as well as calcium signaling pathway, oxidative stress tolerance, and an enzyme involved in de novo L-serine synthesis in the brain treated with A .” I think some words are missing at the end of the sentence.

The quality of English language is OK.

Author Response

8 March, 2023

Nutrients

Dear Editor:

Please find enclosed our revised manuscript. Following to reviewer’s comment, we revised the title from “Orally administrating Tyr-Trp dipeptide antagonizes aberrant gene expression in the mouse cerebral cortex treated with β-amyloid peptide fragments 25–35” to “Bioinformatic analysis of microarray data showing that Tyr-Trp dipeptide antagonizes aberrant gene expression in the mouse cerebral cortex treated with β-amyloid peptide fragment 25–35”.

We thank the editor and reviewers for the detailed and helpful comments. We revised the manuscript accordingly as indicated in our point-by-point responses detailed. We hope that you will find the revised manuscript suitable for publication in Nutrients.

Sincerely,

Momoko Hamano

Assistant professor

Department of Bioscience and Bioinformatics

Faculty of Computer Science and Systems Engineering

Kyushu Institute of Technology

Phone/FAX : +81-948-29-7831

E-mail : momoko@bio.kyutech.ac.jp

Shigeki Furuya

Professor

Laboratory of Functional Genomics and Metabolism

Graduate School of Bioresource and Bioenvironmental Sciences,

Kyushu University

Tel: +81-92-802-4741; Fax: +81-92-802-4741

E-mail: furuya.shigeki.805@m.kyushu-u.ac.jp

The author performed multiple bioinformatics analysis on the microarray data of A 25-35/YW-treated mouse brain, as well as inferred the molecular network of YW action in the brain. I am afraid I have to reject this manuscript at this time because I can not see any Figures in this manuscript so I can’t decide. And the supplementary files also do not include the figures. For these context I can see, please check the abstract, line the second last sentence of the abstract, “We found that YW can not only reverse inflammation-related responses but also activate diverse molecular networks involving transcriptional regulatory system mediated by CREBBP (CBP), EGR family, ELK1, and PPAR, as well as calcium signaling pathway, oxidative stress tolerance, and an enzyme involved in de novo L-serine synthesis in the brain treated with A .” I think some words are missing at the end of the sentence.

Response:

We sincerely apologize for the incompleteness of the submitted manuscript. When we submitted not only the manuscript but also the figure and table PDF files at the same time. However, the figure and tables were not inserted in the manuscript prepared by the journal. In addition, ‘β’ was partly garbled, so the ‘β’ was missing from the text in the abstract you point out. We have inserted the figures and tables into the manuscript and improved the garbled ‘β’, so please check the modified manuscript.

The section of the abstract you pointed out has been revised as follows:

We found that YW can not only reverse inflammation-related responses but also activate diverse molecular networks involving transcriptional regulatory system mediated by CREBBP (CBP), EGR family, ELK1, and PPAR, as well as calcium signaling pathway, oxidative stress tolerance, and an enzyme involved in de novo L-serine synthesis in the brain treated with Aβ25–35.

Round 2

Reviewer 1 Report

Thank you for the clarification on some of the comments. I would like to add a few more comments:

Line 271: "We elucidated the molecular mechanism of the improving short-term 271 memory impairment in the mouse model of AD through YW administration" needs to be backed up by functional studies. Bioinformatics analysis tells us about the possible gene modulations, however, to confirm whether those genes are actually involved in the pathway involving YW would require knockout/knock-in functional studies to support your claim. 

For the updated LC-MS method, it might be beneficial to include the MS spectra of the YW standard along with the one detected in the soy in your supplementary figure to better support your argument.  

Author Response

Line 271: "We elucidated the molecular mechanism of the improving short-term 271 memory impairment in the mouse model of AD through YW administration" needs to be backed up by functional studies. Bioinformatics analysis tells us about the possible gene modulations, however, to confirm whether those genes are actually involved in the pathway involving YW would require knockout/knock-in functional studies to support your claim. 

Response:

While we understand this point made by the reviewer 1, the present study is intended to deduce the pathways responding to YW treatment from bioinformatics analysis. Functional analysis requires time-consuming experimental conditions to set up in vivo experiments in which multiple genes are either suppressed or overexpressed, which is well beyond the scope of this study. Therefore, the sentence has been added to Line 277 to follow your comment.

Page 6 Line 277

On the basis of these results in this study, the functional analysis of the amelioration of gene expression by YW administration will be needed. In particular, further studies including genetic manipulation are needed to determine the functional involvement of individual genes in pathways responding to YW administration.

For the updated LC-MS method, it might be beneficial to include the MS spectra of the YW standard along with the one detected in the soy in your supplementary figure to better support your argument.  

Response:

As pointed out by reviewer 1, we also understand that showing the LC-MS method and MS spectra is very important for the quality of this paper. Indeed, we have already detected Tyr-Trp from certain food protein-derived peptide mixtures by LC-MS (TI’s unpublished observation: Fig. I, K). Based on these results, we are now planning to investigate the effects of the consumption of peptides derived from food proteins containing high levels of Tyr-Trp on cognitive function in the elderly. We would like to publish the LC-MS method and MS spectra along with the results of this clinical study in the original article. Therefore, we are not disclosing details of the peptide analysis (Table 1 shown below) at present. We appreciate your understanding of our plans.

Unpublished data : FigⅠ. Detection of Tyr-Trp Peak in various peptide preparations from plant proteins.

Every letter shows different plant sources. Values are means ± S.E.M. (n=3).

Table Ⅰ. A part of LC-MS method

Reviewer 2 Report

The paper investigated the mechanism of the protective effect of Tyr-Trp dipeptide on β-amyloid peptide 25-35 induced working memory deficits in mouse by bioinformatic analysis of microarray data. It is a topic of interest to the researchers in the related areas, but the paper needs some improvement before acceptance for publication. My detailed comments are as follows:

1.     The title is a little too long, I suggest the author to revise the title to be shorter.

2.     The introduction part needs to revise. It should mention more research about bioinformatic analysis of microarray data, since it is in the title and the main research tool in this part, so it deserves more space.  

3.     The last sentence of the last paragraph of the introduction, maybe it needs to revise to be distinguished from the sentence in the abstract.

4.     The number of the mouse in each group in this paper is n=4 or n=5, actually it is a little bit fewer, we usually at least n=6.

5.     Fig. 4A and 4B, Fig. 5A, 5B, 5C, Fig. 6A, Fig 6B and Fig 6C are too small to see.

The quality of English language of this paper is good, it only needs minor improvement.

Author Response

  1. The title is a little too long, I suggest the author to revise the title to be shorter.

Response:

Following your comment, we revised the title from “Bioinformatic analysis of microarray data showing that Tyr-Trp dipeptide antagonizes aberrant gene expression in the mouse cerebral cortex treated with β-amyloid peptide fragment 25–35” to “Bioinformatic analysis of the molecular networks associated with amelioration of aberrant gene expression by Tyr-Trp dipeptide in the brain treated with amyloid-β peptide”.

  1. The introduction part needs to revise. It should mention more research about bioinformatic analysis of microarray data, since it is in the title and the main research tool in this part, so it deserves more space.

Response:

As this study show the results analyzed by only in silico approach, the significance of bioinformatic analysis needs to be fully explained in introduction. Following your pointed out, one paragraph has been added to the introduction as below.

Line 70 Page 2

Bioinformatics is one of the powerful research fields for analyzing vast and complex omics data, and bioinformatic approaches have been used to elucidate complex molecular mechanisms in our previous studies (Hamano et al., Data in Brief (2016); Hamano et al., Nutrients (2021)). In central nervous system, microarray analysis has clarified the molecular mechanisms of neurogenesis defects by Phgdh deletion (Kawakami et al., Neurosci. Res. (2009)) and the detection of crucial hub genes network following molecular pathological mechanisms (Takashima and Hamano et al., Int J Clin Oncol. (2023)). In addition, comprehensive analyses of gene expression data have been enabled to evaluate the effect on brain function in response to chemicals and nutritional conditions (Nakamura et al. J.Agric. Food Chem (2010); Okada et al., BioFactors (2012)). Based on these findings, bioinformatic analysis of brain gene expression data is useful to elucidate the molecular mechanism of effect of food peptide administration on the brain.

  1. The last sentence of the last paragraph of the introduction, maybe it needs to revise to be distinguished from the sentence in the abstract.

Response:

We agree with the reviewer’s comment that the last sentence of Introduction was similar to that in abstract. We have revised the last sentence in introduction as below.

Line 72 Page 2

Interestingly, we show here that YW can drive diverse molecular networks involving transcriptional regulatory system mediated by calcium signaling pathway as well as oxidative stress tolerance in addition to certain transcription factors. Further, the present analysis implies that YW administration ameliorated the expression level of CREBBP (CBP) and an enzyme involved in de novo L-serine synthesis in Aβ25–35-treated brain.

  1. The number of the mouse in each group in this paper is n=4 or n=5, actually it is a little bit fewer, we usually at least n=6.

Response:

As the reviewer 2 pointed out, it may be necessary to prepare more mice for the analysis in consideration of individual differences of mice. However, we have demonstrated in several previous studies that the number of samples in the present study is sufficient to evaluate the brain metabolism and gene expression alteration induced by the administration of food ingredients (Ichinose et al. (2015), Ichinose et al. (2020)). Following our previous study, we set the number of samples in this study.

  1. Fig. 4A and 4B, Fig. 5A, 5B, 5C, Fig. 6A, Fig 6B and Fig 6C are too small to see.

Response:

As you point out, the relevant figures in the manuscript have been enlarged.
